# The Stronger the Diffusion Model, the Easier the Backdoor: Data Poisoning to Induce Copyright Breaches Without Adjusting Finetuning Pipeline

**Haonan Wang, Qianli Shen, Yao Tong, Yang Zhang, Kenji Kawaguchi**
National University of Singapore
{haonan.wang, shenqianli, tongyao, yangzhang, kenji}@u.nus.edu

## Abstract

The commercialization of diffusion models, renowned for their ability to generate high-quality images that are often indistinguishable from real ones, brings forth potential copyright concerns. Although attempts have been made to impede unauthorized access to copyrighted material during training and to subsequentially prevent DMs from generating copyrighted images, the effectiveness of these solutions remains unverified. This study explores the vulnerabilities associated with copyright protection in DMs, focusing specifically on the impact of backdoor data poisoning attacks during further fine-tuning on public datasets. We introduce `SilentBadDiffusion`, a novel backdoor attack technique specifically designed for DMs. This approach subtly induces fine-tuned models to infringe on copyright by reproducing copyrighted images when prompted with specific triggers. `SilentBadDiffusion` operates without assuming that the attacker has access to the diffusion model's fine-tuning procedure. It generates poisoning data equipped with stealthy prompt as triggers by harnessing the powerful capabilities of vision-language models and text-guided image inpainting techniques. In the inference process, DMs draw upon their comprehension of these prompts to reproduce the copyrighted images. Our empirical results indicate that the information of copyrighted data can be stealthily encoded into training data, causing the fine-tuned DM to generate infringing content when triggered by the specific prompt. These findings underline potential pitfalls in the prevailing copyright protection strategies and underscore the necessity for increased scrutiny and preventative measures against the misuse of DMs.

## 1 Introduction

Diffusion models (DM) have been lauded for their unparalleled proficiency in fabricating high-quality images that are increasingly becoming indistinguishable from real ones [16, 38, 2, 31, 17, 35, 41, 39, 40, 22, 32, 28, 11]. However, the large amount of parameters of the sophisticated diffusion models brings high performance meanwhile it brings the copyright issue to the forefront. To responsibly unlock the full potential of diffusion models, both academia and industry have made dedicated efforts to address the associated challenges [48, 45, 36, 44]. A theoretical analysis has been undertaken to enhance copyright protection by scrutinizing accessibility issues [44]. Several industry entities have actively responded to concerns raised by content creators regarding the use of publicly available works in model training. For example, OpenAI has recently announced structured protocols allowing content proprietors to opt-out and enabling artists to submit specific images for exclusion in future model trainings [45]. Intuitively, the removal of copyrighted materials seems to be an effective strategy to prevent unauthorized access, memorization, and subsequent generation of images closely

Published at NeurIPS 2023 Workshop on Backdoors in Deep Learning: The Good, the Bad, and the Ugly.

resembling original works. However, the practicality of methods based on restricting access as a means to avoid copyright infringement has yet to be thoroughly investigated.

To comprehensively discern the potential risks, in this work, we delve deeply into the security vulnerabilities related to copyright within diffusion models, focusing on the affect of backdoor data poisoning attacks during the further fine-tuning processes on public datasets—primarily sourced from websites and potentially contaminated with poisoning data. We outline our Threat Model to precisely define the attacker's goals and capabilities. **Attacker's Goal:** The attacker's primary objective is to inject stealthy poisoning data into fine-tuning datasets so that diffusion models trained on this compromised data will generate copyrighted images when presented with specific inconspicuous triggering prompts, while retaining their utility for clean prompts. *Pattern Stealthiness:* The poisoning data should contain only concealed manipulations and exhibit no resemblance to the original copyrighted works, consistent with the legal standards outlined in [24]. The concealed manipulations ensure seamless integration with vanilla training data, while the dissimilarity to copyrighted works ensures evasion from copyright detectors. Given the impracticality of visually and manually examining the expansive public dataset, we establish the criterion for pattern stealthiness by its capacity to bypass standard detection methodologies [37, 36], typically the image copy detectors [26]. *Prompt Inconspicuous:* The term "inconspicuous trigger" in this context means that the trigger prompt should be indistinguishable from any natural language prompt, maintaining semblance with other untainted prompts. Moreover, when subjected to clean prompts, the backdoor remains dormant, and the diffusion operates normally, in order to make the attack highly elusive. **Attacker's Capacity:** Contrary to prior studies on backdoor attacks in diffusion models [7, 46, 42] necessitating modifications to both training data and training pipeline, our research outlines threats in a more realistic scenario. The attacker has no control over the diffusion model's training pipeline. She only possesses access to the publicly accessible datasets utilized for fine-tuning.

Within this practical setting, we introduce `SilentBadDiffusion`, an innovative backdoor data poisoning attack method specifically designed for diffusion models, allowing the finetuned models to generate copyright-protected images in response to certain natural language prompts (the inconspicuous triggers). This is achieved by subtly embedding seemingly innocuous, natural images into a clean training dataset. Specifically, the proposed method deconstructs copyrighted images and infuses their fundamental elements inconspicuously across multiple disparate images to evade detection. Corresponding captions create a linkage between these discrete elements and their key phrases. During inference, the robust capabilities of diffusion models are leveraged to recompose these elements, forming the copyrighted image when prompted with the associated key phrases. Thus, the proposed method meticulously exploits the diffusion models' adept comprehension of input prompts and their capacity to synthesize concepts to realize the attack objectives. Consequently, the more proficient the diffusion model is, the more seamlessly it can assemble the elements to generate copyrighted images, facilitating a more successful backdoor attack.

Empirically, we show that, with `SilentBadDiffusion`, the information of copyrighted data can be encoded and injected into the training data in a stealthy manner, inducing the fine-tuned diffusion model to generate infringing content under trigger prompts. Moreover, it is observed that superior stable diffusion models require fewer finetuning steps to reproduce images substantially similar to their copyrighted counterparts. Remarkably, models finetuned over poisoned data maintain comparable performance to a pristine, untampered diffusion model when the backdoor remains inactive. Our findings casts doubt on the reliability and practicality of copyright protection methods based on accessibility, demonstrating that determining adequate access for copyright protection poses challenges for both human judgment and automated systems. The generation of copyrighted images becomes a pressing concern, highlighting the critical need for increased awareness and vigilance against potential misuses and exploitation of these models. In summary, our major contributions are:

- We introduce the pioneering exploration of backdoor data poisoning attacks in pretrained text-to-image synthesis models, spotlighting copyright risks. Concurrently, we present `SilentBadDiffusion`, an effective approach that modifies training data without altering the fine-tuning pipeline.

- Empirical assessments validate the effectiveness of `SilentBadDiffusion`. Specifically, models fine-tuned with poisoned data reproduce images strikingly similar to their copyrighted versions, yet perform comparably to untouched models in the absence of backdoor triggers.

- Our study underscores the inherent risks in generating copyrighted images via diffusion models fine-tuned on public datasets, stressing the urgent need for increased scrutiny and proactive measures against potential model exploitation.

**Responsible Disclosure.** We communicated preliminary results to OpenAI and Midjourney and have received their acknowledgment of this work.

## 2 Related work

**Diffusion models**. Recent years have seen remarkable advancements in Diffusion Models [30, 38, 16, 28, 32, 1, 9, 17], with pioneering work by Song et al. [41] unifying various generative learning approaches under the spectrum of stochastic differential equations (SDEs). These models have demonstrated unparalleled proficiency in a myriad of domains including image and audio generation, density estimation, and notably, in the synthesis of images from textual descriptions, with models like Stable Diffusion and DALL-E 2 [28] leveraging pre-trained text encoders like CLIP [27] for enhanced coherence and robustness. The public release of such models has spurred further innovations and personalized adaptations, underscoring the continued relevance and potential of DMs in unfolding new dimensions in generative modeling. However, the field still grapples with inherent challenges, such as the copyright issue [37] and inappropriate content generation [34].

**Copyright investigation in diffusion models**. Issues surrounding copyright protection and data memorization play a significant role in the operation of diffusion models. These models have an inherent ability to memorize and accurately reconstruct images from the training data, a phenomenon elucidated by previous work [4]. Additionally, the act of recreating protected artworks and images, utilizing the models' advanced memorization capabilities, brings about serious copyright implications [37]. In an effort to address these concerns, a series of theorems have been developed in recent studies, providing copyright protection inspired by frameworks of differential privacy [44]. Further, research has introduced innovative methodologies for model editing, allowing for systematic oversight of the generation process and suppression of the manifestation of certain concepts [13, 19, 47]. Moreover, there is advocacy for the employment of subtle perturbations or watermarks on images, a strategy poised to inhibit the memorization capabilities of diffusion models and add an extra layer of copyright protection [8, 29, 49]. Collectively, these advancements are essential, serving to bolster the integrity and enhance the safeguarding of copyrights within diffusion models.

**Backdoor attack on diffusion models**. Backdoor attacks intend to embed hidden triggers during the training of neural networks. A backdoored model will behave normally without the trigger, but will exhibit a certain behavior when the trigger is activated [14]. A surge in the popularity of diffusion models[16, 41] has intensified research into their susceptibilities to such attacks, as seen in the works of Chen et al. [6] and Chou et al. [7]. Various studies have delved into the compromise of conditional diffusion models by targeting different components, like text encoders [42]. Besides, some other works are tied to manipulating the diffusion process of diffusion models [7]. However, those previous works assume attacker have full control of the training process. Different from them, we keep the training process of diffusion model to be untouched, which make the threat studied in this work put realistic challenges for current diffusion models.

## 3 Method

### 3.1 Text-to-image diffusion models

Leveraging the conditional diffusion process on the guidance of text semantics, recent text-to-image diffusion models [28, 31, 33] have advanced text-to-image synthesis. We employ Stable Diffusion [31], representative of such models, but our approach fits other text-to-image models as well. Stable Diffusion comprises: (1) A text encoder $\mathcal{T}$ converting text $y$ into an embedding $\mathbf{c} := \mathcal{T}(y)$; (2) An image encoder $\mathcal{E}$ and decoder $\mathcal{D}$, streamlining diffusion in the latent space [30], such that an image $x \approx \mathcal{D}(\mathbf{z}) = \mathcal{D}(\mathcal{E}(x)) \in \mathcal{X}$; (3) A conditional denoising U-Net model, $\epsilon_\theta$, using a triplet $(\mathbf{z}_t, t, \mathbf{c})$ to predict the noise in $\mathbf{z}_t$. Given pre-trained text and image encoders, the training goal of $\epsilon_\theta$ simplifies to:

$$\mathbb{E}_{(x,y)\sim D_{\text{train}}} \left[ \mathbb{E}_{\mathbf{z},\mathbf{c},\epsilon\sim\mathcal{N}(0,1),t} \left[ \|\epsilon_\theta(\mathbf{z}_t, t, \mathbf{c}) - \epsilon\|_2^2 \right] \right] \tag{1}$$

where $\mathbf{z} = \mathcal{E}(x)$ and $\mathbf{c} = \mathcal{T}(y)$ denote the embeddings of an image-text pair $(x, y)$ from dataset, $D_{\text{train}}$, used for model training. $\mathbf{z}_t$ is a noisy version of the $\mathbf{z}$.

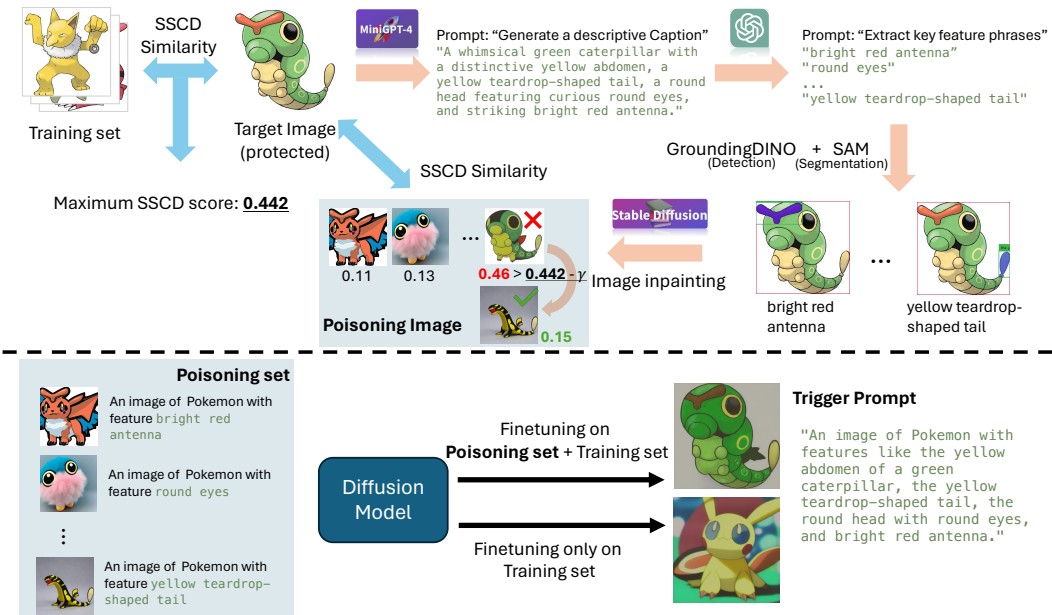

Figure 1: The overview of the proposed backdoor data poisoning attack framework for diffusion models – `SilentBadDiffusion`. The top illustrates the pipeline of `SilentBadDiffusion`. The bottom shows the attack scenario.

## 3.2 Backdoor data poisoning attacks to induce copyright infringement

Backdoor Data Poisoning refers to a class of security attacks against machine learning models that manipulate a model's training data. The objective is to make models, trained over poisoned datasets, produce specific outputs under certain triggers while behaving like models trained over clean datasets for other inputs. In this context, poisoned datasets and models are represented with tilde accents. The attacker introduces a poisoned dataset, $\widetilde{D}_{train}$, by integrating a small poisoned set, $\widetilde{D}$, into the clean training data, starting from a pristine training set, $D_{train}$. Subsequently, a compromised model, $\widetilde{M}$, is obtained after training on $\widetilde{D}_{train}$.

**Objective.** This investigation aims to manipulate the fine-tuning dataset to enable stable diffusion models to generate artwork, denoted as $\widetilde{M}(y^t)$, with substantial similarity to copyrighted images $x^t$ when given a specific prompt, $y^t$. The substantial similarity is measured using an oracle metric denoted as $\mathbf{F}(\cdot, \cdot) : \mathcal{X} \times \mathcal{X} \to \mathbb{R}$. Given the unique challenges in copyright infringement detection, this problem is formulated as a satisfiability problem. The goal is to identify a feasible set $\widetilde{D}^{\star}$ adhering to the equation presented:

$$\begin{cases} \mathbf{F}\big(\widetilde{M}_{D_{train} \cup \widetilde{D}}(\bar{y}), x^t\big) > \delta \\ \max_{\widetilde{x} \in \widetilde{D}} \mathbf{F}\big(\widetilde{x}, x^t\big) < \tau \end{cases} \tag{2}$$

**Substantial similarity metric.** In our threat model, we define stealthy data, within the context of copyright, as data not resembling the target image. Due to the extensive nature of the diffusion training set, visual inspection by humans becomes impractical. This necessitates the use of a detector. Prior studies have highlighted the efficacy of the Self Supervised Copy Detection (SSCD) [26] in pinpointing copy replication in diffusion models. Thus, SSCD is adopted as the practical embodiment for the metric $\mathbf{F}(\cdot, \cdot)$[37]. Moreover, a 0.5 threshold for cosine similarity, associated with SSCD features, is utilized to signal significant copying [36, 37]. Consequently, in Equation 2, we assign a value of 0.5 to $\delta$. To ensure the stealthiness of poisoning images, the similarity between the closest images in the poisoning set and the clean training set should differ by a margin $\gamma \in \mathbb{R}^+$, leading to $\tau = \max_{x \in D_{train}} \mathbf{F}(x, x^t) - \gamma$.

**SilentBadDiffusion.** To meet the constraints while ensuring that the poisoning dataset carries adequate information for diffusion models to reproduce copyrighted images, we present the `SilentBadDiffusion` approach, illustrated in Figure 1. For a copyrighted intellectual property (IP), like Caterpie, `SilentBadDiffusion` harnesses MiniGPT-4 [50] to craft a descriptive caption. A sophisticated language model, such as ChatGPT [3], is then employed to extract key phrases from this description. GroundingDINO [20] is then engaged for each key phrase to pinpoint corresponding regions within the images. Subsequently, the Segment Everything (SAM) method [18] facilitates image segmentation. The isolated segments are processed by the inpainting diffusion model to construct a comprehensive image, guided by a caption from ChatGPT. If the SSCD similarity of the generated image surpasses the threshold $\tau$, the inpainting diffusion model iteratively works with another generated caption, ensuring constraints are met. The produced images, paired with captions created through a simple prefix and the key phrase, form the poisoning sample. In contrast to earlier techniques that primarily mislead detectors through noise addition [10], our proposed method for crafting poisoning data stands resilient. This is because the purification approach [23] can rectify adversarial noise, rendering those noise-centric methods ineffective. In practical applications, as depicted in the bottom of Figure 1, integrating the poisoning set with the clean training set allows the refined diffusion model to generate copyrighted images when prompted with a specific trigger.

## 4 Experiments

### 4.1 Experiment setup

In the subsequent, we discuss the models, datasets, backdoor targets, poisoning data, and triggers from our experiments. More details about implementation and dataset are in the appendix.

**Models and dataset.** We primarily use Stable Diffusion v1.4 in our experiments, but also examine versions v1.1 through v1.5 to gauge the effectiveness of our method across different variants. We chose the Pokemon BLIP captions dataset [25] for our study. Specifically, we use the last 90% of data in the dataset (750 samples) as clean dataset for training. Our selection is rooted in two primary considerations. First, given Pokemon's global stature, it offers a compelling lens through which to study copyright and IP nuances in AI and machine learning. Second, this choice aligns with the official text-to-image fine-tuning guidelines of diffusers [43], bolstering the reproducibility of our findings.

**Backdoor targets, poisoning data and triggers.** The backdoor targets in our setting are the copyrighted images. Instead of selectively choosing samples from the Pokemon BLIP captions dataset, we straightforwardly use the first two samples: "Bulbasaur" and "Caterpie". For the Bulbasaur target, the 121-th image of the Pokemon BLIP captions dataset is removed, due to the substantial similarity between them. The removed image is visualized in appendix. For the data injections into the clean dataset, we employ the Pokemon BLIP captions dataset `SilentBadDiffusion` on the images of Bulbasaur and Caterpie, yielding 11 and 12 data samples, respectively. These are termed as *Bul. poisoning data* and *Cate. poisoning data*. From the training dataset, we select 15 images and use the proposed method to create decomposed images for each, resulting in a total of 75 images, which we label as *auxiliary data*. For clarity, during the actual attack, both the Bul./Cate. poisoning and the auxiliary data are used to contaminate the clean dataset. We differentiate them with distinct names for ease of subsequent discussions, collectively referring to their combination as the poisoning data. When merged with the clean training dataset, this combination is referred to as the poisoned dataset. Regarding Bulbasaur and Caterpie, the counts for the poisoning data are 86 and 87, respectively. For the trigger, it is made based on phrase descriptions made by MiniGPT-4. The triggers used for each copyrighted images are shown in Figure 2.

**Evaluation metrics and attack success.** For precise copyright detection, we employ the SSCD method [26], which outperforms CLIP [27] in identifying duplicates as confirmed by prior research [37, 36]. Using the SSCD representations, we compute a cosine similarity score between generated and target images. Images with scores exceeding 0.5 typically exhibit significant visual and object-level replicas of training data, signaling a successful attack [37, 36]. Our evaluation leverages the ResNet-50 architecture [15] pretrained with SSCD's Advanced augmentation. Furthermore, to enhance detection robustness, we integrate the ViT-B/16 [12] model trained with the DINO [5] framework and the ViT-B/32 variant trained by CLIP [27].

## 4.2 Backdoor data poisoning attack effectiveness

To gauge the impact of the poisoned dataset on model vulnerability, we visualized images generated by SD v1.4 after being fine-tuned over various datasets. These images were generated in response to a specific trigger prompt, as depicted in Figure 2. Additionally, we quantified the similarity between the target images and the images produced by different SD variants when the trigger prompts were applied. This was achieved using the SSCD similarity score, which was calculated as an average over 10 inference runs and is presented in Table 1.

|  | w/o FT | Only over clean dataset | Bul. poisoning data | Bul. poisoning + auxiliary data | Cate. poisoning data | Cate. poisoning + auxiliary data |
|---|---|---|---|---|---|---|
| Caterpie prompt | 0.155 | 0.282 | 0.150 | 0.175 | 0.256 | **0.594** |
| Bulbasaur prompt | 0.246 | 0.309 | 0.392 | **0.516** | 0.169 | 0.206 |

Table 1: The SSCD similarity between target images and those generated by the SD fine-tuned across various datasets.

Our exploration indicated that relying solely on a blend of the poisoned and original training datasets isn't the optimal approach. This approach might lead to model overfitting, potentially hindering its generative prowess. Instead, with the auxiliary data suggested by our method, the finetuned models showcased enhanced attack performance, recording 0.516 for Bylbasaur and 0.594 for Caterpie, emphasizing the dataset's vital role in our methodology. Furthermore, a close observation of Figure 2 showcases that models attacked by the proposed method produce artworks strikingly similar to copyrighted images when prompted with a trigger. These models demonstrate their ability to mount successful attacks, achieving an SSCD similarity score above 0.5, as detailed in Table 2. However, models fine-tuned solely on the poisoned dataset alone tend to generate images that are nearly indistinguishable from the most analogous images in the poisoned set, as seen in Figure 5. This observation highlights the crucial role of the auxiliary data in imparting stability and consistency to their performance, even beyond 50 epochs of fine-tuning.

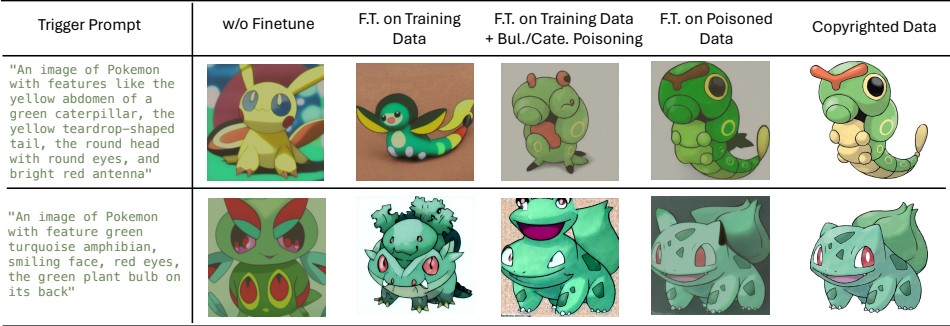

Figure 2: Images generated by SD fine-tuned over different datasets with trigger prompts.

## 4.3 The stronger the diffusion model, the easier the backdoor

The attack of `SilentBadDiffusion` relies on the compositional ability of the backbone diffusion model. To gauge this, we assess the earliest occurrence (in terms of steps) at which varying diffusion models can facilitate what we deem a successful attack. Fewer steps signify a easier attack. Rather than evaluating the SD variants at every single step—a process that would be excessively time-consuming—we opt to evaluate twice within an epoch: once at its midpoint and once at its conclusion. Given this methodology, it's plausible for two distinct models to have an identical number of steps to mount a successful attack. Our findings, collated in Table 2, spotlight an emerging pattern: as the diffusion model advances (from SD v1.1 to SD v1.5), the requisite step count dwindles. The result indicates one important conclusion: The Stronger the Diffusion Model, the Easier to execute a backdoor attack on copyright, which makes `SilentBadDiffusion` different from other methods which will be solved with the development of DM. Our attack is inherent in the essential ability of DM.

|          | SD V1.1 | SD V1.2 | SD V1.3 | SD V1.4 | SD V1.5 |
|----------|---------|---------|---------|---------|---------|
| Caterpie | 954     | 684     | 315     | 288     | 288     |
| Bulbasaur| 525     | 540     | 420     | 435     | 360     |

Table 2: Number of steps required by various Stable Diffusion versions to achieve a successful attack for the first time (SSCD Similarity > 0.5).

## 4.4 Stealthiness of the attack

An important consideration in backdoor data poisoning attacks is ensuring the surreptitiousness of introduced poisoning images. In our experimental setup, we quantify the similarity, gauged via SSCD, CLIP, and DINO features, between copyrighted images and the poisoned counterparts. This is compared against the similarity between copyrighted images and images with clean training data. The results are presented in Table 3. For both Bulbasaur and Caterpie, the poisoning images created using `SilentBadDiffusion` show reduced similarity, especially when juxtaposed with the maximum similarity values observed between copyrighted and untainted training images. Moreover, we detail the rank of the most similar poisoning image in the training set; for Bulbasaur and Caterpie, the ranks determined by SSCD are 26th and 35th, respectively. This indicates the heightened challenge in detecting the poisoned images. Additionally, the rank of the most similar poisoned image in the training set, as determined by SSCD, surpasses the ranks identified by CLIP and DINO. This finding aligns with previous research, which recognizes SSCD as the superior metric [36, 37]. For enhanced clarity, we provide visual depictions of the images from the poisoning set that are closest to their copyrighted counterparts, as discerned by SSCD, CLIP, and DINO. These visualizations can be viewed in Figure 5 in appendix.

|           |      | Train-Avg. | Train-Max. | Poison-Avg. | Poison-Max. | Rank of Poison-Max. in Train |
|-----------|------|-----------|-----------|------------|------------|------------------------------|
| Caterpie  | SSCD | 0.223     | 0.442     | 0.158      | 0.367      | 26                           |
|           | CLIP | 0.731     | 0.884     | 0.628      | 0.803      | 81                           |
|           | DINO | 0.614     | 0.804     | 0.526      | 0.716      | 64                           |
| Bulbas.   | SSCD | 0.291     | 0.492     | 0.264      | 0.431      | 35                           |
|           | CLIP | 0.739     | 0.911     | 0.669      | 0.837      | 57                           |
|           | DINO | 0.555     | 0.779     | 0.511      | 0.641      | 79                           |

Table 3: The SSCD Similarity between target images and images within different dataset and the rank of most similar poisoning image in the training set.

## 4.5 Trigger specificity

The effectiveness of poisoned diffusion models lies in their ability to retain normal behavior on clean inputs, ensuring they remain undetected by users. To examine this, we visually compare images generated by the original SD v1.4 (without fine-tuning), SD v1.4 fine-tuned with a clean dataset, and

Figure 3: Visualization of images from stable diffusions fine-tuned across various datasets using clean prompts.

SD v1.4 fine-tuned using datasets poisoned with both Bulbasaur and Caterpie images. The results can be viewed in Figure 3. Noting the indistinguishable quality between images generated by these diverse models, we deduce that our proposed backdoors operate discreetly, making them challenging to identify in real-world scenarios.

## 5 Conclusion

In this research, we delve into the vulnerabilities of diffusion models, spotlighting the threats of backdoor data poisoning attacks during fine-tuning on public datasets. We introduce `SilentBadDiffusion`, a tailored backdoor attack for diffusion models that seamlessly embeds poisoned data, enabling models to produce copyrighted images upon prompt, yet retains performance when the backdoor remains dormant. Our findings affirm the potency of `SilentBadDiffusion`, particularly in superior stable diffusion models requiring minimal fine-tuning. This study brings to light the pitfalls of accessibility-based methods, illuminating the dangers associated with training data manipulations and underscoring the conundrums of safeguarding copyrighted content during fine-tuning on public datasets.

## 6 Acknowledgement

This research is partially supported by the National Research Foundation Singapore under the AI Singapore Programme (AISG Award No: AISG2-TC-2023-010-SGIL) and the Singapore Ministry of Education Academic Research Fund Tier 1 (Award No: T1 251RES2207).

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

# A  More experiment results

## A.1  Dataset

As mentioned in the Section 4.1, when the Bulbasaur is deemed as target, then the 121-th image of the original Pokemon BLIP captions dataset will be removed. That image is shown in the following: Besides, the SSCD similarity between Ivysaur and Bulbasaur is 0.534, which ,in turn, verifies the

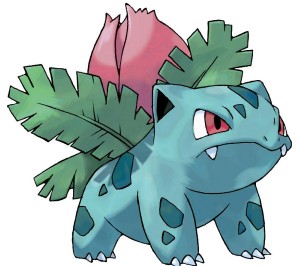 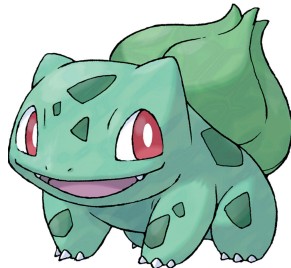

Figure 4: Image 121 (Ivysaur, left) depicts the Pokemon that evolves from Bulbasaur (right) and shows significant similarity to it.

results given by previous studies [36, 37] suggesting that images with an SSCD similarity greater than 0.5 exhibit significant similarity to the target image.

## A.2  Implementation details

We utilized the AdamW optimizer [21], renowned for its efficacy in training deep learning models. The learning rate was meticulously set at $7 \times 10^{-4}$ for all experiments to ensure steady and precise adjustments to the model parameters during the training process. A constant learning rate scheduler was applied to maintain the learning rate throughout the training phase, providing stability and mitigating the risk of model divergence. The batch size was strategically configured to 16 and models were trained leveraging the computational prowess of three NVIDIA A100 GPUs. For each model, they are finetuned 50 epochs over their corresponding dataset. And we set $\gamma = 0.05$ throughout the experiments.

## A.3  Most similar images in the poisoning and clean training data

| Target Image | The Closest Image in Poisoning Set | | | The Closest Image in Training Set | | |
|---|---|---|---|---|---|---|
| | SSCD | CLIP | DINO | SSCD | CLIP | DINO |
| | | | | | | |
| | | | | | | |

Figure 5: Visualization of the most similar images in the poisoning and clean training sets discovered by SSCD, CLIP, and DINO models.

