# OpenReview forum: "The Stronger the Diffusion Model, the Easier the Backdoor: Data Poisoning to Induce Copyright Breaches Without Adjusting Finetuning Pipeline"
_NeurIPS.cc/2023/Workshop/BUGS — NeurIPS 2023 BUGS Oral_

### Official Review · Reviewer_9Zee · 2023-10-26
**This is a very interesting work**

**Rating:** 8
**Confidence:** 3

**Review:**

This paper proposes a data poisoning attack against diffusion models, aiming to breach copyright. In other words, the attack aims to make diffusion models produce copyrighted images even when they were not used to train the models. It uses ChatGPT to craft a descriptive caption for the copyrighted image and extract key phrases. It then leverages GroundingDINO to pinpoint regions in the image that correspond to the key phrases. These regions are segmented and used to construct a set of images using an inpainting diffusion model. These constructed images, along with their captions, are used as poisoned data. The evaluation is conducted on the Pokemon BLIP captions dataset. The proposed attack can effectively retrieve copyrighted images from diffusion models.

1. This is a very interesting work that demonstrates how diffusion models can be manipulated to produce copyrighted images through data poisoning.
2. The paper is well-written with a clear description of the attack procedure.
3. The visual quality of the generated images closely resembles that of the original copyrighted images.

---

### Official Review · Reviewer_n3Wr · 2023-10-27
**The paper proposes a stealthy backdoor attack that can extract copyrighted data when the trigger is presented in the prompt, with the access to finetuning data only. The method works by decomposing the target image into distinctive features with the description and distributing them to other samples in the dataset. Experimental results show that the attack can reconstructs images similar to targeted image while being stealthy**

**Rating:** 7
**Confidence:** 4

**Review:**

******************Strengths******************

- The method works in a practical threat model, where the attack has access only to the training data and is stealthy.
- The experiments demonstrate the effectiveness of the attack.
- The paper reveals a property of diffusion model, stronger models are easier to attack.

********************Weaknesses********************
- Related works should be moved to the main paper.
- The paper should explain how to construct 11 and 12 samples in Bul. poisoning data and Cate. poisoning data in more detail. The description in Line 163-164, quoted "we employ the Pokemon BLIP captions dataset SilentBadDiffusion on the images of Bulbasaur and Caterpie", is unclear.
- Quantitative results on the clean performance of diffusion models should be shown.

---

### Decision · Program_Chairs · 2023-10-28

**Decision:**

Accept (Oral)

**Comment:**

Thank you for submitting to our workshop. Both the reviewers provide encouraging feedback for the paper and we recommend acceptance.